# Construction and Validation of a Novel Prognosis Model in Colon Cancer Based on Cuproptosis-Related Long Non-Coding RNAs

**DOI:** 10.3390/jcm12041528

**Published:** 2023-02-15

**Authors:** Guan-Zhan Liang, Xiao-Feng Wen, Yi-Wen Song, Zong-Jin Zhang, Jing Chen, Yong-Le Chen, Wei-Dong Pan, Xiao-Wen He, Tuo Hu, Zhen-Yu Xian

**Affiliations:** 1Department of Colorectal Surgery, Department of General Surgery, Guangdong Provincial Key Laboratory of Colorectal and Pelvic Floor Diseases, The Sixth Affiliated Hospital, Sun Yat-sen University, Guangzhou 510655, China; 2Department of Radiotherapy, The Sixth Affiliated Hospital, Sun Yat-sen University, Guangzhou 510655, China; 3Department of Pancreatic Hepatobiliary Surgery, Department of General Surgery, The Sixth Affiliated Hospital, Sun Yat-sen University, Guangzhou 510655, China

**Keywords:** colon cancer, cuproptosis, long non-coding RNA, prognosis model, targeted therapy

## Abstract

Colon cancer (CC) is one of the most common (6%) malignancies and leading cause of cancer-associated death (more than 0.5 million) worldwide, which demands reliable prognostic biomarkers. Cuproptosis is a novel modality of regulated cell death triggered by the accumulation of intracellular copper. LncRNAs have been reported as prognostic signatures in different types of tumors. However, the correlation between cuproptosis-related lncRNAs (CRLs) and CC remains unclear. Data of CC patients were downloaded from public databases. The prognosis-associated CRLs were identified by co-expression analysis and univariate Cox. Least absolute shrinkage and selection operator were utilized to construct the CRLs-based prognostic signature in silico for CC patients. CRLs level was validated in human CC cell lines and patient tissues. ROC curve and Kaplan–Meier curve results revealed that high CRLs-risk score was associated with poor prognosis in CC patients. Moreover, the nomogram revealed that this model possessed a steady prognostic prediction capability with C-index as 0.68. More importantly, CC patients with high CRLs-risk score were more sensitive to eight targeted therapy drugs. The prognostic prediction power of the CRLs-risk score was further confirmed by cell lines, tissues and two independent CC cohorts. This study constructed a novel ten-CRLs-based prognosis model for CC patients. The CRLs-risk score is expected to serve as a promising prognostic biomarker and predict targeted therapy response in CC patients.

## 1. Introduction

Colon cancer (CC) is one of the most commonly diagnosed digestive cancers worldwide, which leads to more than 0.5 million deaths every year [1,2]. Although the incidence and mortality of CC have gradually decreased in some developed countries with the rapid development of diagnostic and therapeutic methods, the CC cases are still increasing in the developing world, including China [3,4,5]. Generally, 20–25% of CC patients are presented with distant metastasis and, approximately another 25% with localized tumors, also develop metastasis at later time point, leading to a 5-year overall survival (OS) rate as low as 14.1% [6,7,8,9]. Despite the advancement of our understanding about CC pathogenesis, there is still lack of effective prognostic biomarkers [10]. For example, CEA, CA125 and CA19-9, the most commonly used tumor biomarkers in clinical practice, are exhibited with relatively low sensitivity and specificity [11,12]. On the other hand, although some new biomarkers such as circulating tumor DNA (ctDNA) have been approved in the prognosis, diagnosis and management of CC, they are also too expensive to be widely applied in clinical practice [13,14]. Therefore, developing a reliable and practical prognostic signature is extremely important for the survival of CC patients.

Long non-coding RNAs (lncRNAs) are a type of non-coding RNA with a length longer than 200 nucleotides [15]. In recent years, numerous studies have shown that lncRNAs play important roles in tumorigenesis and metastasis [16]. For example, previous researches demonstrated that lncRNAs could promote the progression of different cancer types including CC [17,18]. Moreover, lncRNAs have been reported to be associated with programmed cell death such as apoptosis and ferroptosis [19,20]. Given their roles in the progression of cancers as well as the characteristic being easily detected in body fluids like blood and urine, lncRNAs could be served as valuable biomarkers and therapeutic targets for cancers [17,21,22]. In many studies, lncRNAs have also been utilized to construct prognostic models for various cancer types, including liver cancer and lung cancer [23,24].

Regulated cell death (RCD)—including apoptosis, necroptosis and ferroptosis—plays important roles in inflammatory responses and tumor therapy [25]. Different from traditional RCD relying on reactive oxygen species (ROS), cuproptosis is a novel modality of mitochondrial RCD triggered by intracellular copper accumulation [26]. Copper can bind to the lipoylated components of tricarboxylic acid cycle and lead to proteotoxic stress [27]. Moreover, elevated copper levels, which may affect the development of cancer, have been found in tumor tissues and serum of cancer patients [28]. Therefore, using copper ionophores to induce cuproptosis could be a potential therapeutic strategy for cancer treatment [29]. The role of cuproptosis-related lncRNAs (CRLs) in CC progression has been not well explored. Therefore, it is very important to identify the key CRLs with prognostic significance in CC patients.

In this study, we identified and validated a ten-CRLs-based prognostic model in CC patients based on data from TCGA. This model can be potentially applied in prognostic prediction and target therapy guidance in CC patients.

## 2. Materials and Methods

### 2.1. Data Collection

The RNA-Seq dataset, mutational data and relevant clinical information of COAD were downloaded from the TCGA database (https://portal.gdc. cancer. gov/, accessed on 24 April 2022) up to 20 April 2022. There were 514 samples in this dataset, including 473 tumor samples and 41 normal tissue samples. A total of 458 tumor samples from 446 CC patients were enrolled in this study after excluding the duplicate samples, non-tumor samples and samples without survival information. Meanwhile, we obtained progression-free survival (PFS) data of TCGA-COAD from UCSC Xena (https://xenabrowser.net/, accessed on 24 April 2022, Dataset ID: Appendix A). The RNA-Seq dataset and relevant clinical information of CC patients from another independent validation cohort were obtained from GEO database (GSE152430).

### 2.2. Validation of Cuproptosis-Related LncRNAs

Cuproptosis-related genes (CRGs) were acquired from the study of Tsvetkov et al. [26], and cuprotosis-related mRNA and lncRNA expression profiles were extracted from TCGA database. The “limma” R package (Version 3.50.3; R Foundation for Statistical Computing, Vienna, Austria) was utilized to filter the cuprotosis-related mRNA and lncRNA expression value of CC (row mean > 0.1) [30]. Then the CRLs were obtained by Pearson’s correlation analysis (|cor| > 0.4, *p* < 0.001).

### 2.3. Construction of Risk Score Model

Univariate Cox regression was applied to identify OS-related CRLs with *p* < 0.01 for model construction. Then, least absolute shrinkage and selection operator (LASSO; “glmnet” R package, version 4.1.4) and multivariate Cox regression analysis were performed to identify the best prognostic signature based on these lncRNAs with *p* < 0.01 in univariate Cox screen analysis [31]. The risk score was calculated using the following formula:*Risk score* = ∑ (*ExplncRNAi* × *CoeflncRNAi*)


*ExplncRNAi* is the expression value of lncRNA, and *CoeflncRNAi* is the estimated regression coefficient of indicated lncRNA in multivariable Cox regression analysis (see Figure 1). On the basis of this, patients in TCGA-COAD were classified into high CRLs-risk group and low CRLs-risk group with median risk score as cutoff.

### 2.4. Validation of Risk Score Model

“Survival” R package (version 3.3.1) was used to perform the Kaplan–Meier survival curve analysis, which preliminarily tested the predictive ability of CRLs-based risk model [32]. The “limma” R package was utilized to conduct Pearson’s correlation analysis between 19 CRGs and 10 CRLs involved in the risk model. Moreover, univariate and multivariate Cox regression analyses were performed to evaluate the association between prognosis and CRLs-risk score as well as clinical characteristics. ROC curve analysis and Kaplan–Meier survival curve analysis were performed to validate the predictive power of this model [33]. Moreover, a nomogram based on CRLs signature and clinical characteristics was established to predict survival of CC patients (see Figure 1). The predictive accuracy of CRLs-risk model was assessed using consistency index (C-index) and calibration curves [34]. In addition, principal component analysis (PCA) was used to explore the distribution of samples in different risk groups based on the expression of whole genome, cuproptosis-related mRNAs, CRLs and prognosis-associated CRLs (“stats” R package, version 4.1.2) [35].

### 2.5. Gene Set Variation Analysis

Gene set variation analysis (GSVA) was implemented to explore the differences of immune function between CC patients in high CRLs-risk group and those in low CRLs-risk group. Single-sample gene set enrichment analysis (ssGSEA) was performed based on GSEA (https://www.gsea-msigdb.org/gsea/index.jsp, accessed on 24 April 2022) via “GSVA” R package (version 1.42.0) [36]. The scores of ssGSEA between above two risk groups were compared by Wilcoxon signed rank test and the significantly enriched immune functions were defined as *p* value < 0.05. The results were presented with a heatmap.

### 2.6. Tumor Mutational Burden Analysis

The genome sequencing data (including somatic mutation data) of COAD was downloaded from the TCGA database, and perl script was used to calculate the gene mutation load tumor mutation burden (TMB; dividing the total number of non-synonymous mutations by the size of the coding region of the target region) [37]. To validate the relationship between CRLs-based prognosis model and colon cancer mutation, the TMB of CC patients in high CRLs-risk and low CRLs-risk groups were compared based on the data from TCGA. The “survival” R package was utilized to perform the Kaplan–Meier survival curve analysis, which illustrated the impact of high TMB and low TMB on overall survival (OS) time of CC patients. Furthermore, survival analysis was conducted for CC patients from different subgroups based on risk score and TMB level.

### 2.7. The Targeted Therapy Drug Sensitivity Analysis

The “pRRophetic” R package (https://github.com/paulgeeleher/pRRophetic, accessed on 24 April 2022, version 0.5) was utilized to evaluate the association between CRLs-risk score and 50% inhibiting concentration (IC_50_) level of targeted therapy drugs [38,39,40,41]. The related IC_50_ data were from the study by Garnett et al. [42]. Then, the targeted therapy drugs with their IC_50_ levels significantly correlated with risk model were presented (*p* < 0.001). Moreover, Spearman’s correlation analysis was performed to further explore the difference of targeted therapy drugs sensitivity between CC patients in above two risk groups, and the results with statistical significance were visualized (*p* < 0.001). 

### 2.8. Cell Culture

The human intestinal epithelial cell line HIEC-6 and CRC cell lines (HCT8, HCT116, SW48, RKO, WiDr) were obtained from American Type Culture Collection (ATCC). HIEC-6, SW48 and Widr were used in the sixth generation, RKO, HCT8 and HCT116 were used in the eighth generation. All cell lines were cultured in a 60 mm TC-treated disposable Petri dish (430166, Corning^®^, Corning, NY, USA). These cells were cultured with Dulbecco’s Modified Eagle’s Medium (DMEM) or RPMI 1640 containing 10% fetal bovine serum and 1% penicillin-streptomycin in 5% CO^2^-humidified incubator under 37 °C. A mycoplasma contamination test was conducted for these cells every four months.

### 2.9. RNA Extraction and Quantitative Polymerase Chain Reaction (qPCR)

To further evaluate the accuracy of CRLs-risk score model, we tested expression levels of ten prognosis-associated CRLs in human CC cell lines as well as paired CC and normal tissue samples from our institution. This study was approved by the ethics committee of the Sixth Affiliated Hospital of Sun Yat-sen University (IRB: 2020ZSLYEC-064, 20 April 2020). TRIzol (15596018, Invitrogen, Carlsbad, CA, USA) was utilized to collect total RNA samples from CC cell lines and paired CC and normal tissue specimens. ReverTra Ace quantitative polymerase chain reaction (qPCR) RT Kit (FSQ-101, Toyobo, Osaka, Japan) was used to perform the reverse transcription reaction. SYBR Green (QP002, ESscience, Shanghai, China) was used as fluorescence. The quantitative PCR was conducted 40 cycles using 10 ng/μL cDNA by ABI QuantStudio™ 7 Flex Real Time PCR Systems. The expression level of prognosis-associated CRLs was normalized by β-actin using the 2^−ΔΔCt^ method. The primers of indicated CRL were listed in Appendix A.

## 3. Results

### 3.1. Identification of CRLs in CC and Construction of CRLs-Based Prognostic Model

The flow chart of methodology utilized for our study is shown in Figure 1. Based on the data from TCGA-COAD, a total of 16,876 lncRNAs were identified. Altogether, 19 CRGs were acquired from the study of Tsvetkov et al. [26], and we extracted the expression profile of CRGs in TCGA-COAD dataset (Appendix A). A total of 2568 CRLs were identified based on the correlation analysis between colon cancer lncRNAs (CCLs) and CRGs (|cor| > 0.4 and *p* < 0.001). Sankey diagram was established to demonstrate the relationship between CRLs and CRGs in CC patients (Figure 2A). Moreover, 60 CRLs were found to be significantly associated with prognosis of CC patients using univariate Cox regression analysis (*p* < 0.05; Figure 2B). Then LASSO regression analysis was performed to filter the overfitting CRLs, which identified 10 prognosis-associated CRLs (LINC02257, AL031985.3, FARSA-AS1, LINC01762, PDE9A-AS1, NSMCE1-DT, AC002066.1, AC104964.3, AC010789.2 and AC092375.2) by multivariate Cox regression analysis (Figure 2C,D). A novel CRLs-based prognostic model was constructed using the following risk score formula (Table 1): 

Risk score = LINC02257 × 0.52885 + AL031985.3 × (−0.50835) + FARSA-AS1 × (−0.72032) + LINC01762 × (−1.59252) + PDE9A-AS1 × (−1.51676) + AC002066.1 × 0.77106 + AC104964.3 × (−0.71065) + NSMCE1-DT × 2.49784 + AC010789.2 × 2.31471 + AC092375.2 × (−0.71840).

To further explore the distinction of the expression profile between CC patients in high CRLs-risk group and low CRLs-risk group, PCA was performed to illustrate the distribution of the whole genome expression (Appendix A), cuproptosis-related mRNAs (Appendix A), CRLs (Appendix A) and prognosis-associated CRLs (Appendix A) in CC patients. These results preliminarily demonstrated that CC patients in high/low CRLs-risk group were exhibited with different cuproptosis states.

### 3.2. Evaluation of the CRLs-Based Prognostic Model in CC Patients

CC patients in the TCGA-COAD dataset were classified into a high CRLs-risk group and low CRLs-risk group with cutoff as median risk score (Figure 3A). The average survival time of CC patients with high CRLs-risk score was much shorter than those with low CRLs-risk score (Figure 3B). Moreover, the expression levels of 10 prognosis-associated CRLs were presented in the heatmap (Figure 3C). Patients in high CRLs-risk group had high expression of LINC02257, AC002066.1, NSMCE1-DT and AC010789.2 but low expression of AL031985.3, FARSA-AS1, LINC01762, PDE9A-AS1, AC104964.3, and AC092375.2. Kaplan–Meier survival curve analysis revealed that the OS time of CC patients in high CRLs-risk group was significantly shorter than those in low CRLs-risk group (*p* < 0.001; Figure 3D). Similarly, the PFS time of patients in high CRLs-risk group was found to be remarkably less than those with low CRLs-risk score (*p* = 0.008; Figure 3E). In addition, survival analysis results illustrated that CRLs-risk score exhibited a better prognostic prediction power in stage I-II CC patients than stage III and stage IV patients (*p* < 0.001 vs. *p* = 0.012 and *p* = 0.010, Figure 3F,H).

The heatmap demonstrated the correlation between expression level of CRLs and CRGs in CC patients (Figure 4A). Moreover, univariate and multivariate Cox regression analysis were performed to evaluate the association between prognosis and CRLs-risk score as well as clinical characteristics in CC patients. The regression analysis results revealed that CRLs-risk score (*p* < 0.001), age (*p* < 0.001), and tumor stage (*p* = 0.004) were independent prognostic indictors for CC patients (Figure 4B,C). Interestingly, the CRLs-risk score in CC patients with lymphatic invasion (LI) was significantly higher than those without LI, similar result was also observed in CC patients with venous invasion (VI) (Figure 4D,E).

To further validate the predictive power of CRLs-risk score, ROC analysis was performed based on the risk model and clinical characteristics. The result showed that CRLs-risk score was an independent risk factor for CC prognosis with a favorable predictive power (Figure 5A). More importantly, the AUC value of CRLs-risk score (0.712) was higher than that of age (0.621), tumor stage (0.617) and gender (0.480) (Figure 5B). The nomogram was developed to predict OS rate of CC patients based on CRLs-risk score and clinical characteristics, with the C-index as 0.680 (Figure 5C). Moreover, the calibration curves indicated that the predictive power of nomogram was quite accurate in CC patients (Figure 5D).

### 3.3. Correlations between CRLs-Risk Score Model and Immune Function, Tumor Mutation Burden in CC Patients

The gene set variation analysis (GSVA) was performed to explore relationship between immune function and CRLs-risk score model. As shown in Figure 6A, the functions of killer T-cell activity, immune checkpoint and para inflammation were statistically higher in CC patients from high CRLs-risk group (*p* < 0.05). To identify the difference of CC-related gene mutations between patients in high and low risk group, the mutation percentage of indicated genes in each group was counted and demonstrated in oncoplots. More CC patients in high CRLs-risk group exhibited gene mutation (98.11% vs. 93.07%, Figure 6B,C). As illustrated in the violin plot, the tumor mutation burden (TMB) was significantly higher in CC patients from high CRLs-risk group (*p* = 0.039; Figure 6D). Furthermore, Kaplan–Meier survival curve analysis revealed that the OS time of patients in the high TMB group was significantly shorter than those in the low TMB group (*p* = 0.021; Figure 6E). Based on these findings, we integrated the CRLs-risk score model and TMB model to predict prognosis of CC patients. Survival analysis results demonstrated that the CRLs-risk score displayed better predictive power in CC patients from low TMB subgroup than those from high TMB subgroup (*p* < 0.001 vs. *p* = 0.064; Figure 6F).

### 3.4. Correlations between CRLs-Risk Score Model and Targeted Therapy Drug Sensitivity

In order to explore the correlation between CRLs-based prognostic signature and targeted therapy drug sensitivity, the IC50 levels of 251 targeted therapy drugs were estimated and nine drugs were significantly correlated with CRLs-risk score (*p* < 0.001). The IC50 levels of 17-AAG, AUY992, GSK-650394, HG-6-64-1, Cisplatin, CCT018159, LFM-A13 and KIN001-055 were founded to be negatively correlated with the risk score, while the IC50 level of Lisitinib was positively associated with the risk score (Figure 7A–I). 

To further identify the difference of targeted therapy drug sensitivity between CC patients in the high risk group and low risk group, the estimated IC50 levels of targeted therapy drug between above two groups were compared. As shown in the boxplots, the IC50 levels of eight drugs (17-AAG, AUY992, GSK-650394, HG-6-64-1, Cisplatin, CCT018159, LFM-A13 and KIN001-055) were significantly lower in patients from the high risk group (*p* < 0.001; Figure 8A–I). These drugs could be the candidate therapy for CC patients with high CRLs-risk score, while Lisitinib might be suitable for those with a low CRLs-risk score. This work supplied some practical guidance for the individualized precision therapy in CC patients.

### 3.5. Validation for the Expression Level and Predictive Power of CRLs-Based Risk Score in CC Cell Lines and Tissues

To further evaluate the function of these 10 prognosis-associated CRLs, we assessed their expression level in human CC cell lines and tissues via qPCR. As demonstrated in Figure 9A, the expression levels of LINC02257, AC002066.1, NSMCE1-DT, and AC010789.2 were significantly higher in CC cell lines (RKO, HT8, HCT116, SW48, WiDr) than those in the normal human intestinal epithelial cell (HIEC-6). On the contrary, the expression levels of AL031985.3, FARSA-AS1, LINC01762, PDE9A-AS1, AC104964.3, and AC092375.2 were remarkably lower in CC cell lines comparing with HIEC-6. Furthermore, the expression levels of 10 prognosis-associated CRLs were evaluated based on 20 paired CC and peritumoral tissue samples from our institution. A similar expression pattern was obtained by qPCR analysis results, LINC02257, AC002066.1, NSMCE1-DT, and AC010789.2 were exhibited with higher expression in CC tissue samples. Conversely, AL031985.3, FARSA-AS1, LINC01762, PDE9A-AS1, AC104964.3, and AC092375.2 showed lower expression levels in tumor tissues than the corresponding matched-peritumoral samples (Figure 9B). These results further confirmed the accuracy and constancy of above bioinformatic analysis based on public database.

To further validate the predictive power of CRLs-risk score, we evaluated the expression levels of 10 prognosis-associated CRLs in CC patients from our institution. The CRLs-risk score was obtained based on expression levels of these 10 prognosis-associated CRLs using formula: *Risk score* = ∑ (*ExplncRNAi* × *CoeflncRNAi*). As shown in Figure 9C, the OS time of CC patients with high CRLs-risk was significantly shorter than those with low CRLs-risk (*p* = 0.008). Moreover, the survival analysis of CC patients from the GSE152430 cohort demonstrated a similar result (*p* = 0.008; Figure 9D). The prognostic prediction power of CRLs-risk score was confirmed by the above two independent CC patients cohort, which further indicated the splendid prediction efficiency of CRLs-risk score.

## 4. Discussion

Colon cancer is one of the most common (6%) malignancies worldwide in terms of both new cases (more than 1.14 million) as well as deaths (more than 0.57 million), which seriously endangers human health [2]. Although the detection and treatment approach have been greatly improved, the 5-year survival rate is only around 64% for CC patients and this data is less than 20% for CC patients with distant metastasis [7,43]. Therefore, identifying the stable and effective prognostic predictors is warranted to improve the survival of CC patients. In this study, we constructed a CRLs-based prognostic model in CC patients using COAD data-set from TCGA. The subgroup survival analysis illustrated that CRLs-risk score exhibited a better prognostic prediction power in stage I-II CC patients than stage III and stage IV patients. These 10 CRLs may play an important role during the onset and early development of CC, which can explain the above phenomenon. The underlying molecular mechanisms for these CRLs in regulating CC onset and early development need to be explored in our future work. The performance of this CRLs-risk score was evaluated and validated by ROC curve analysis and calibration curves. The expression levels of 10 prognosis-associated CRLs were assessed in human CC cell lines and tissues from our institution via qPCR. The CRLs signature was demonstrated with powerful and accurate prediction power in TCGA-COAD cohort, which could better guide the clinical management of CC patients. The prognostic prediction efficiency of the CRLs-risk score was further validated by two independent CC patient cohorts. This CRLs-based risk score could accurately predict prognosis and provide a theoretical basis for targeted therapy selection of CC patients.

Compared with CRLs signature in other cancer types, such as five-CRLs signature (FOXD2-AS1, NRAV, MED8-AS1, WARS2-AS1 and MKLN1-AS) in hepatocellular carcinoma and three-CRLs signature (AC026401.3, FOXD2-AS1, and LASTR) in clear cell renal cell carcinoma [44,45], our study constructed a prognostic signature based on 10 CRLs (LINC02257, AC002066.1, NSMCE1-DT, AC010789.2, AL031985.3, FARSA-AS1, LINC01762, PDE9A-AS1, AC104964.3 and AC092375.2) for CC patients. Since the collinearity between gene expressions is more obvious than the clinical characteristics, it is necessary to eliminate the interaction effects with appropriate statistical methods. Therefore, we used LASSO to remove the interaction effect, which is more suitable for RNA-Seq data [46]. Some studies have reported the role and function of above CRLs in different types of cancers. Consistent with our study, several previous studies demonstrated that LINC02257 was a prognostic marker for patients with colon adenocarcinoma or renal clear cell carcinoma. In detail, high expression of LINC02257 in tumor tissues is associated with poor prognosis by regulating PI3K/AKT signaling pathway [47,48]. Furthermore, Zhou et al. [49] showed that LncRNA FARSA-AS1 promoted proliferation, stemness, and metastasis of colorectal cancer cells through upregulating FARSA and SOX9, which was different from our study. In addition, NSMCE1-DT, AL031985.3 and LINC01762 were reported to be involved in the prognosis of colon adenocarcinoma, hepatocellular carcinoma and renal cell carcinoma, respectively [50,51,52]. However, the role of these CRLs in the progression of CC remains unclear and further research is necessary.

The GVSA result revealed that expression levels of genes involved in cytolytic activity, checkpoint and para inflammation pathway were significantly upregulated in CC patients from high CRLs-risk group. This may indicate the important role of immune dysfunction in cuproptosis and prognosis of CC patients. Therefore, more investigations are warranted to clarify the relationship and specific mechanisms between cuproptosis and immune response. 

TMB is not only a predictor for prognosis, but also an indicator for immunotherapy efficacy in cancer patients [53]. In general, cancer patients with high TMB could obtain more benefits from immunotherapy as neoantigens derived from gene mutation might induce higher immunogenicity [54,55,56]. However, there is still debate about the effect of TMB on the prognosis of tumor patients, including CC patients [55,56]. Dung et al. [57] demonstrated that CRC patients with high TMB might gain more benefit from immunotherapy, while Zhou et al. [58] revealed that CC patients with low TMB exhibited higher OS rate than those with high TMB. Consistent with Zhou et al., our result suggested that CC patients with low TMB obtained longer OS and the CRLs-risk score displayed better predictive power in patients from low TMB subgroup. Previous studies demonstrated that lncRNAs could affect TMB and thereby regulate immunotherapy response in multiple cancer types [59,60]. Therefore, CRLs signature may affect the prognosis of tumor patients by regulating their TMB status, which needs to be further explored in future studies. 

In the past decades, targeted therapy has evolved rapidly and is widely utilized in different types of tumors including CC [61,62]. For example, targeting JAK2/STAT3 signaling pathway has been utilized in breast cancer treatment [63]. Moreover, metastatic CRC patients with wild-type Kras could benefit from cetuximab treatment [64]. However, numerous targeted therapy drugs did not exhibit satisfactory results in CC patients, because of its heterogeneity [65]. Our study revealed the correlation between CRLs-risk score and sensitivity of 9 targeted therapy drugs. CC patients with high CRLs-risk score might respond better to some certain targeted therapy drugs such as 17-AAG, AUY992, GSK-650394, HG-6-64-1, Cisplatin, CCT018159, LFM-A13 and KIN001-055. However, none of above agents has been utilized in colon cancer patients. In addition, the IC50 level differences between the low/high CRLs-risk groups are relatively small. Therefore, these results need to be carefully interpreted. Whether the CRLs-risk score could serve as guidance for targeted therapy in CC patients demands more meticulous research. Moreover, further studies are required to find out the molecular mechanism underlying CRLs-risk score and targeted drug sensitivity in cancer patients.

In previous studies, some lncRNAs-based prognostic signatures of CC have been reported, such as ferroptosis-related lncRNAs and pyroptosis-related lncRNAs signatures [66,67]. However, until now, there is no CRLs-based prognostic model for CC patients. Based on CRLs, our study constructed and validated a new signature composed of 10 CRLs for CC patients with good predictive power, which could be applied in clinical management of CC patients. Especially, the CRLs-risk score presented better prediction efficacy for CC patients with stage I-II, which indicates that the CRLs-risk signature can be utilized in early screening of CC. Moreover, this CRLs-risk score could provide abundant and comprehensive support for prognostic prediction and treatment selection of CC patients. Additionally, to verify the efficacy of our signature, we evaluated the expression level of these 10 CRLs, whose expression trend was basically consistent with our prediction based on previous bioinformatic analysis. However, there are still some limitations need to be addressed. Firstly, due to the lack of complete expression profiles of lncRNAs and clinical data in public databases, external validation was just performed in a single CC patients cohort from GEO database. Secondly, we only checked the expression level of 10 prognosis-associated CRLs in 5 CC cell lines and 60 CC patients’ tissues. Therefore, further validation based on more clinical samples and other databases would be helpful to provide more solid evidence. In addition, considering heterogeneity of tumor samples might affect the predictive power of CRLs-risk score, we tested its predictive ability in three independent cohorts from TCGA-COAD, GEO (GSE152430), and the Six Affiliated Hospital of Sun Yat-sen University. The CRLs-risk score was demonstrated with good predictive power in above three cohorts. Given tumor heterogeneity existed among different individuals and within tumor tissues, more independent cohorts and advanced technologies such as single cell sequencing are required for further validation work. Furthermore, the underlying molecular mechanism of how these lncRNAs regulate cuproptosis process and CC progression remains unknown. Therefore, further studies are required to explore the exact functions of these CRLs on the development of CC.

## 5. Conclusions

In conclusion, our study constructed and validated a CRLs-based prognostic signature in CC patient cohorts from a public database and our institution. This CRLs-risk score is a promising prognostic predictive biomarker, which can contribute to the development of individualized precision therapy for CC patients.

## Figures and Tables

**Figure 1 jcm-12-01528-f001:**
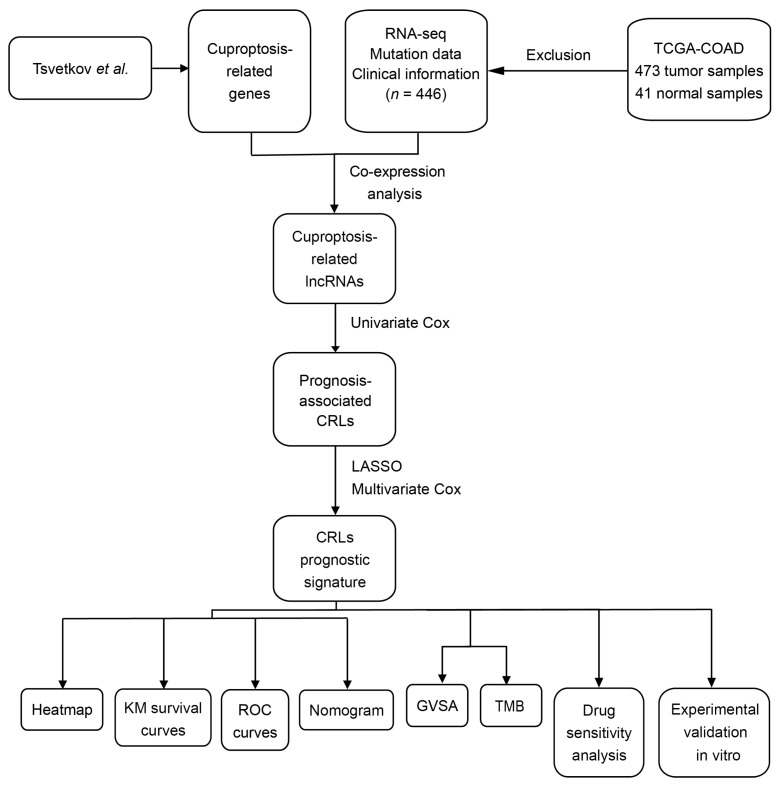
Flow chart of this research.

**Figure 2 jcm-12-01528-f002:**
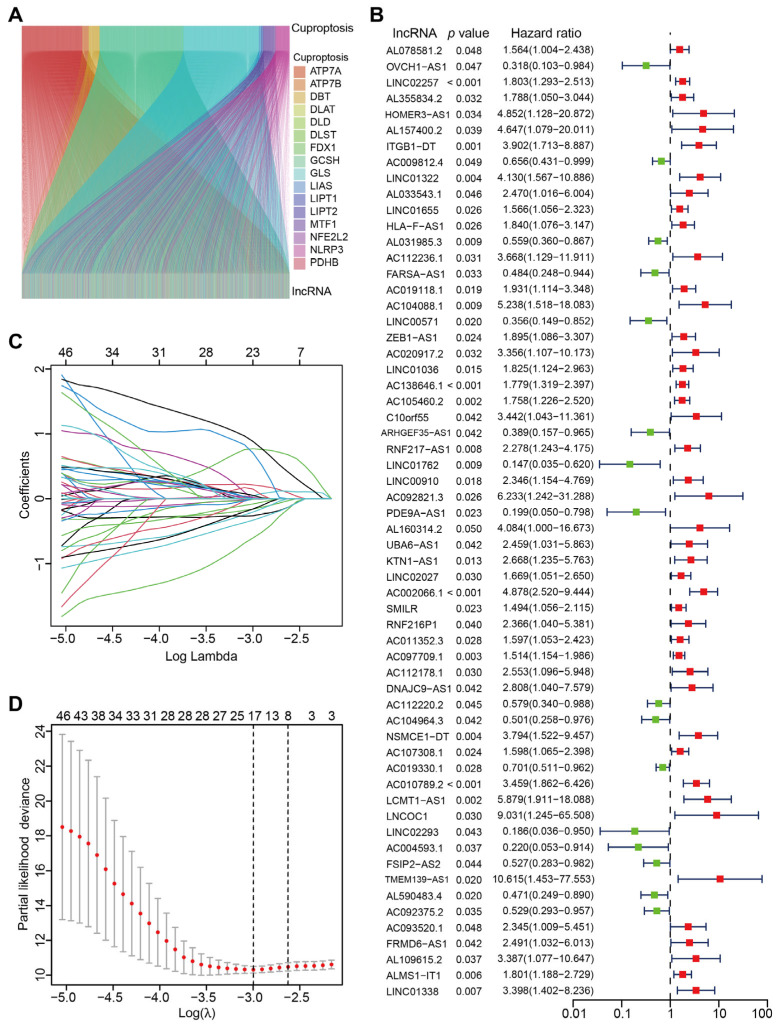
Identification of cuproptosis-prognosis-related LncRNAs. (**A**) Sankey diagram depicting the lncRNAs associated with cuproptosis-related genes (|cor| > 0.4 and *p* < 0.001). (**B**) Forest plot demonstrating 60 CRLs associated with prognosis of CC patients (*p* < 0.05). (**C**) Lasso coefficient of 60 CRLs associated with OS in CC patients. (**D**) The 60 CRLs based on LASSO cross validation plot.

**Figure 3 jcm-12-01528-f003:**
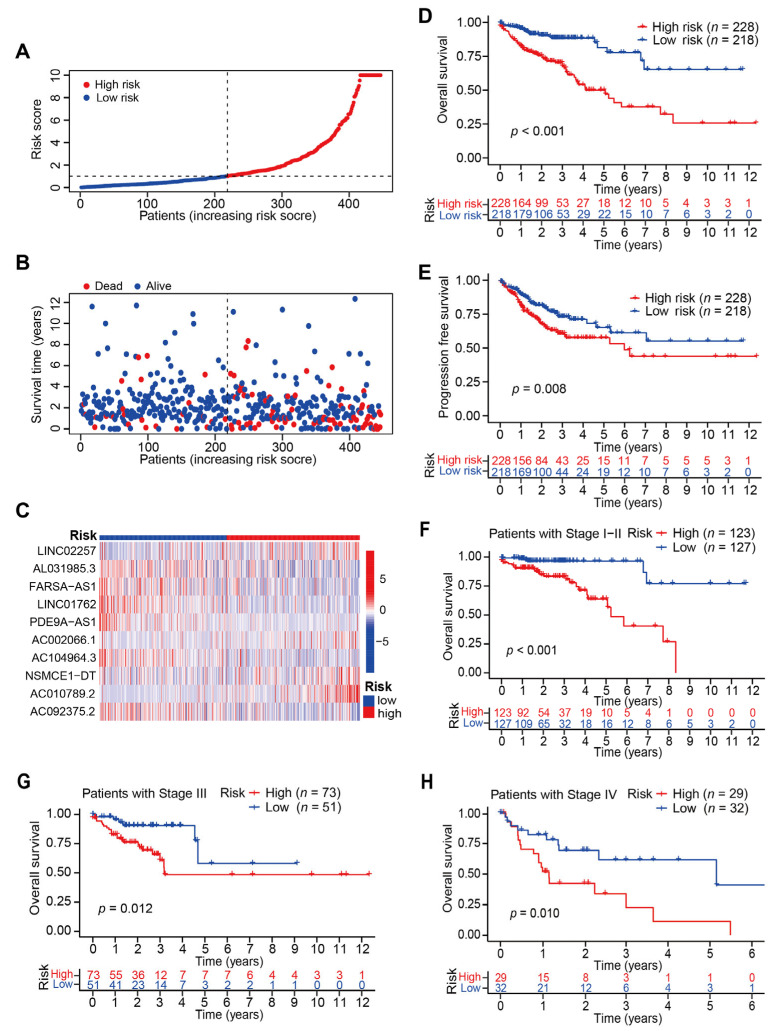
Survival analysis for CC patients based on cuproptosis-prognosis-related LncRNAs. (**A**,**B**) Distribution and survival status of CC patients with different CRLs-risk score. (**C**) Heatmap for the expression profiles of 10 CRLs. (**D**) Kaplan–Meier survival curve of OS in CC patients with high/low CRLs-risk score (*p* < 0.001). (**E**) Kaplan–Meier survival curve of PFS in CC patients with high/low CRLs-risk score (*p* = 0.008). (**F**) Kaplan–Meier survival curve of OS in stage I-II CC patients with high and low CRLs-risk score (*p* < 0.001). (**G**) Kaplan–Meier survival curve of OS in stage III CC patients with high and low CRLs-risk score (*p* = 0.012). (**H**) Kaplan–Meier survival curve of OS in stage IV CC patients with high and low CRLs-risk score (*p* = 0.010).

**Figure 4 jcm-12-01528-f004:**
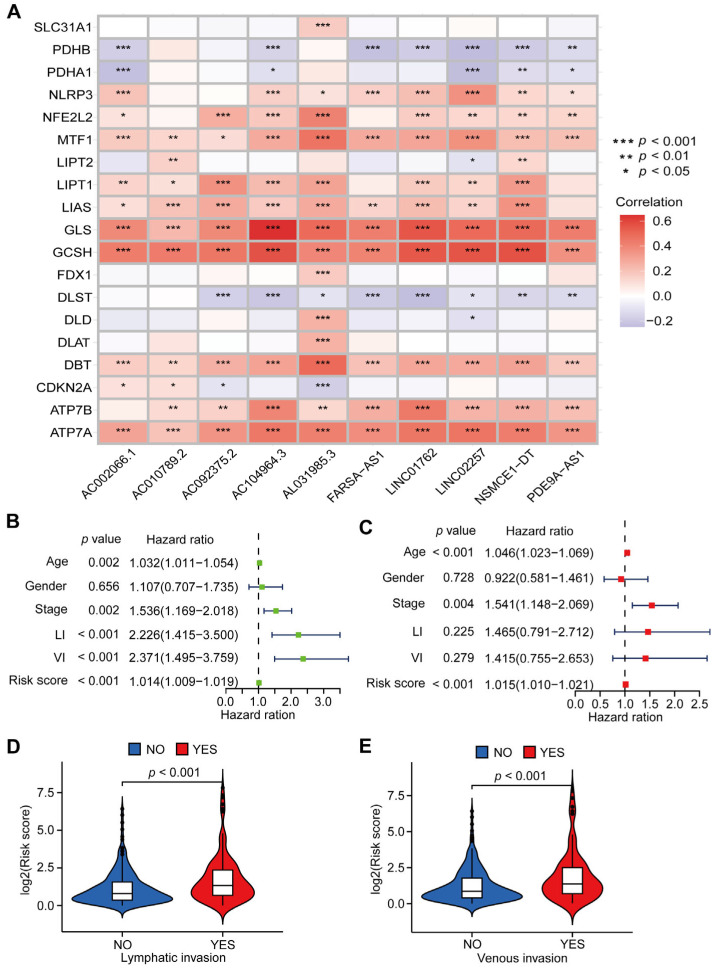
Cox analysis for prognostic characteristics of CRLs. (**A**) The correlation heatmap between 10 prognostic CRLs and 19 cuproptosis-related genes (* *p* < 0.05, ** *p* < 0.01, and *** *p* < 0.001). (**B**,**C**) Univariate Cox regression analysis (**B**) and multivariate Cox regression analysis (**C**) for the CRLs-risk score and clinical characteristics in CC patients (LI means lymphatic invasion and VI means venous invasion). (**D**) Risk score of patients with or without LI (“NO” group means patients without LI and “YES” group means patients with LI, *p* < 0.001). (**E**) Risk score of patients with or without venous invasion (“NO” group means patients without VI and “YES” group means patients with VI, *p* < 0.001).

**Figure 5 jcm-12-01528-f005:**
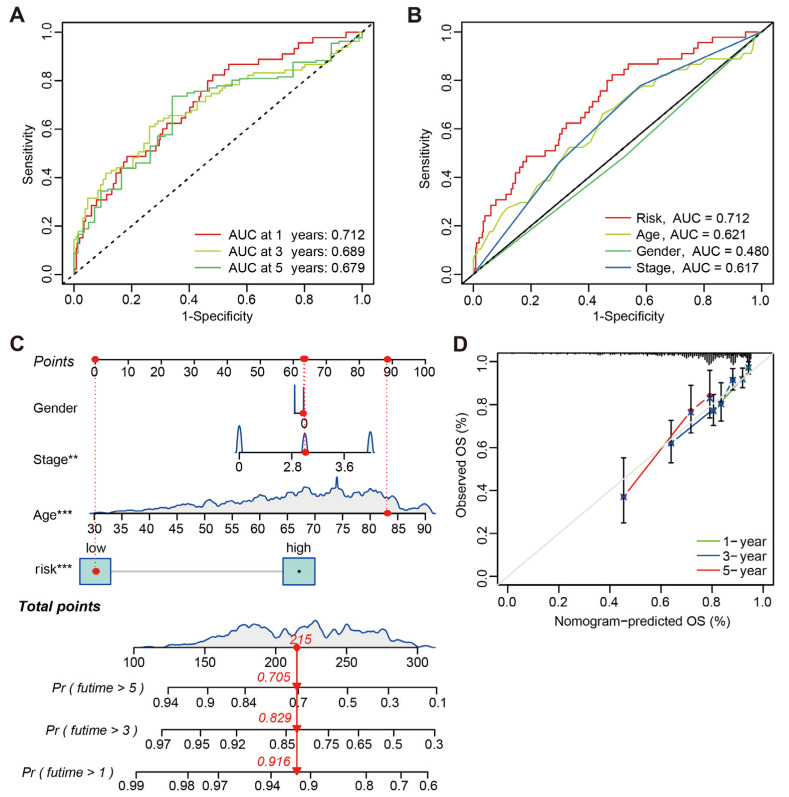
Prognostic prediction efficacy of the CRLs-based risk score model. (**A**) ROC curve analysis result of the CRLs-risk score. (**B**) ROC curve of the CRLs-risk score and clinical characteristics for prognostic prediction power in CC patients. (**C**) Prognostic nomogram based on CRLs-risk score and clinical characteristics to predict survival rates of CC patients (** *p* < 0.01 and *** *p* < 0.001). (**D**) Calibration curves of nomogram between the predicted and observed survival rates in CC patients.

**Figure 6 jcm-12-01528-f006:**
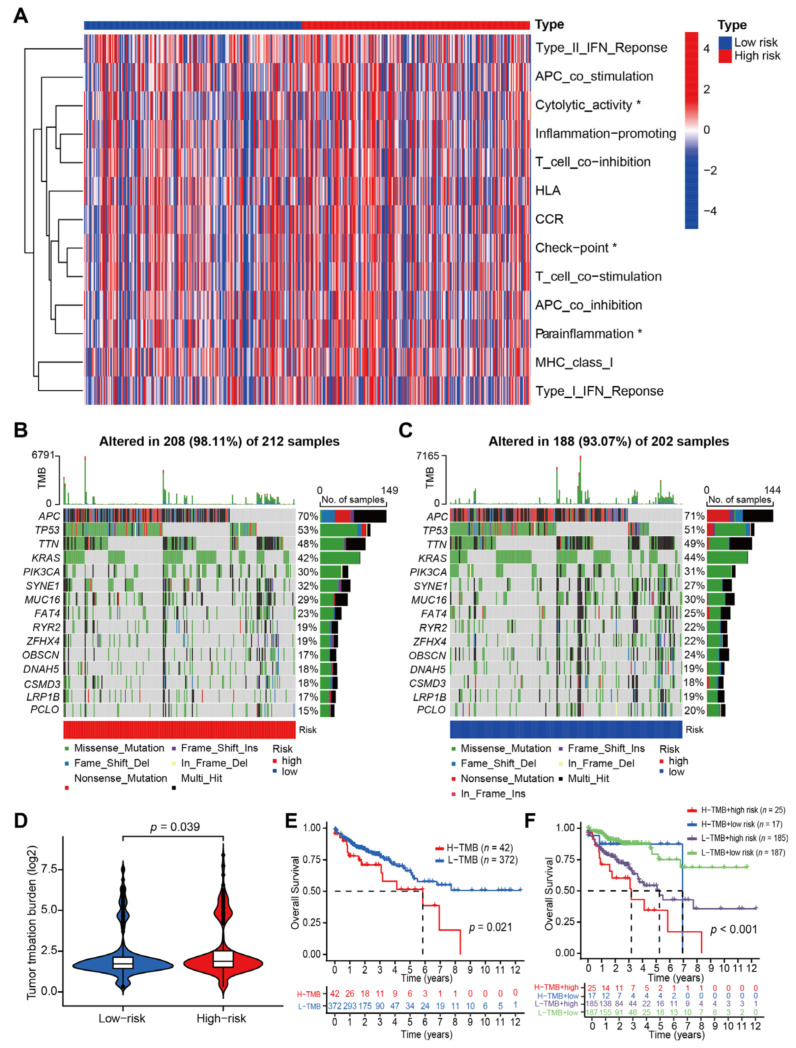
GVSA analysis of prognostic characteristics and TMB analysis. (**A**) Immune pathway enrichment heatmap for prognostic characteristics in CC patients with high and low CRLs-risk score (* *p* < 0.05). (**B**,**C**) Oncoplot demonstrating tumor mutational burden in CC patients with high (**B**) and low (**C**) CRLs-risk score. (**D**) Violin plot for statistical analysis of TMB in high CRLs-risk and low CRLs-risk CC patients (*p* = 0.039). (**E**) Kaplan–Meier survival curve of OS in high TMB and low TMB CC patients (*p* = 0.021). (**F**) Kaplan–Meier survival curve of OS in CC patients with different TMB status and risk score (*p* < 0.001).

**Figure 7 jcm-12-01528-f007:**
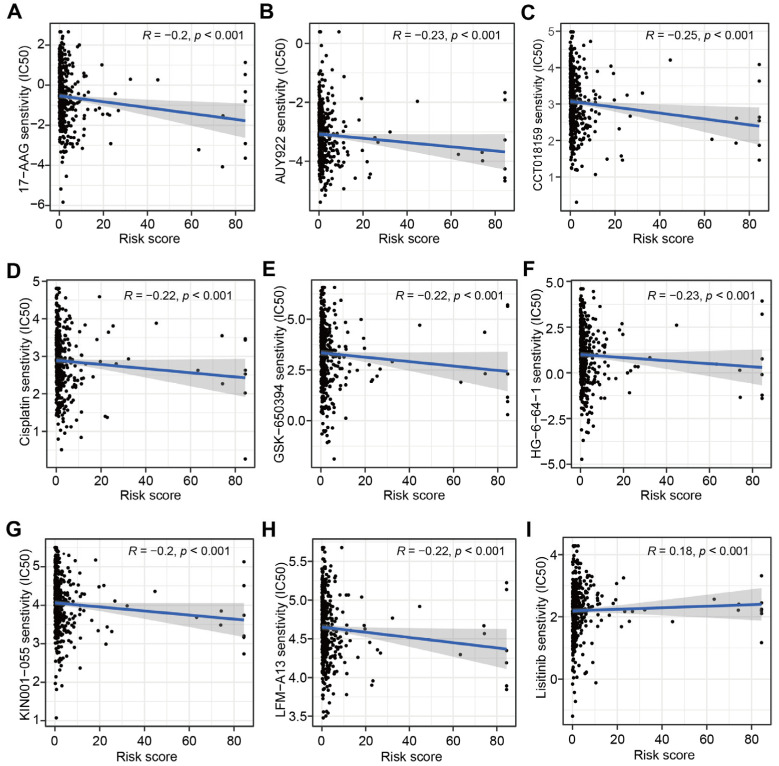
Correlation analysis between sensitivity of targeted therapy drugs and CRLs-based risk score. (**A**–**I**) Correlation analysis between the IC50 value of 17-AAG (**A**), AUY992 (**B**), CCT018159 (**C**), Cisplatin (**D**), GSK-650394 (**E**), HG-6-64-1 (**F**), KIN001-055 (**G**), LFM-A13 (**H**), Lisitinib (**I**) and CRLs-risk score.

**Figure 8 jcm-12-01528-f008:**
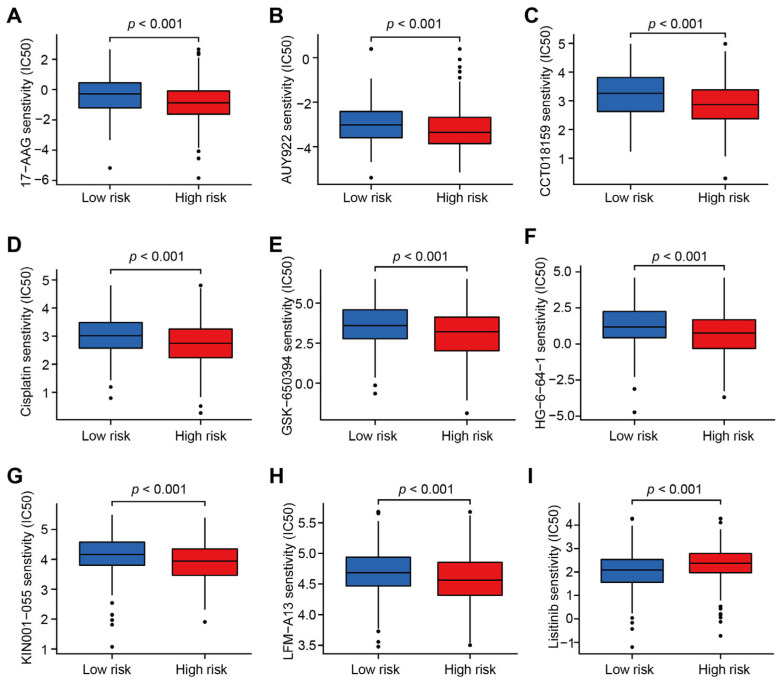
Sensitivity of targeted therapy drugs in CC patients from different risk groups. (**A**–**I**) Box plot depicting the IC50 value of 17-AAG (**A**), AUY992 (**B**), CCT018159 (**C**), Cisplatin (**D**), GSK-650394 (**E**), HG-6-64-1 (**F**), KIN001-055 (**G**), LFM-A13 (**H**), and Lisitinib (**I**) in CC patients from different CRLs-risk groups.

**Figure 9 jcm-12-01528-f009:**
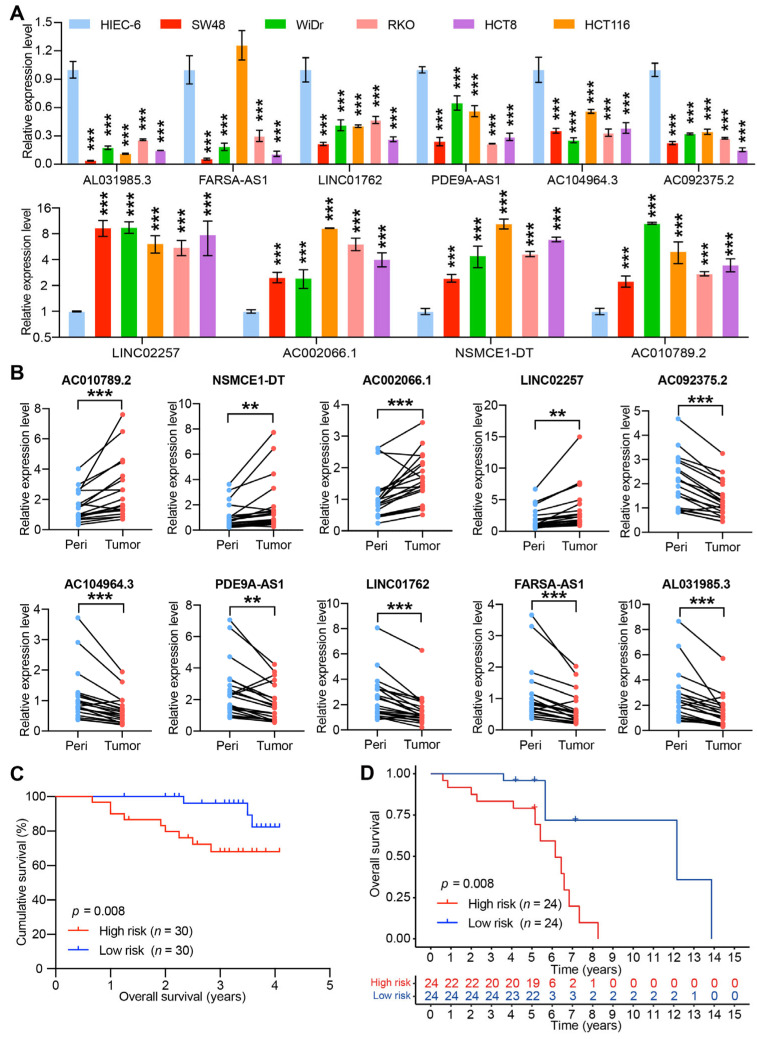
Validation for the expression level and predictive power of CRLs-based risk score in CC cell lines and tissues. (**A**) Expression level of ten prognosis-associated CRLs in five human colon cancer cell lines (RKO, HT8, HCT116, SW48, WiDr) and normal human intestinal epithelial cell (HIEC-6). (**B**) Expression level of ten prognosis-associated CRLs in 20 paired CC and peritumoral tissue samples. (**C**) Survival analysis of CC patients layered by the CRLs-risk score in cohort from the Sixth Affiliated Hospital, Sun Yat-sen University (*p* = 0.008). (**D**) Kaplan–Meier survival curve of OS in CC patients with high/low CRLs-risk score from GSE152430 cohort (** *p* < 0.01 and *** *p* < 0.001).

**Table 1 jcm-12-01528-t001:** Multivariate Cox results for cuproptosis-related long non-coding RNA based on The Cancer Genome Atlas—colon adenocarcinoma (TCGA-COAD).

Id	Coef.	HR	HR.95L	HR.95H	*p*-Value
LINC02257	0.52885	1.80251	1.2929398	2.5129063	0.00051
AL031985.3	−0.50835	0.5587	0.360131289	0.8667413	0.009369
FARSA-AS1	−0.72032	0.4838	0.248071542	0.9435233	0.033127
LINC01762	−1.59252	0.14719	0.034968938	0.6195109	0.008977
PDE9A-AS1	−1.51676	0.19917	0.049698797	0.7982032	0.022715
AC002066.1	0.77106	4.87808	2.519773295	9.4435596	2.58 × 10^−6^
AC104964.3	−0.71065	0.50146	0.257631679	0.9760437	0.042222
NSMCE1-DT	2.49784	3.79419	1.522266783	9.4568597	0.004213
AC010789.2	2.31471	3.45939	1.862392022	6.425818	8.56 × 10^−5^
AC092375.2	−0.71840	0.52911	0.292583435	0.956856	0.035216

## Data Availability

The data are available from the corresponding author.

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
