# Peer review of "Construction and Validation of a Novel Prognosis Model in Colon Cancer Based on Cuproptosis-Related Long Non-Coding RNAs"

_jcm, 2023, doi:10.3390/jcm12041528_

Round 1

Reviewer 1 Report

Journal of Clinical Medicine/ oncology section

Manuscript ID: jcm-2086346

Title 

Authors have constructed in silico their prognosis model; Their title must include that the model was constructed in silico and validated in cell lines and patient tissues.

Abstract

1.     The abstract's first sentence, "Colon cancer (CC) is one of the most common malignancies, which demand reliable prognostic biomarkers” must be more informative and scientifically address the health issues caused by colorectal cancer.

2.      The different parts of the abstract must be equiprobable in size, the methods part is longer than the others.

3.     This sentence is part of the study methods and must be moved to the methods section, not between results and conclusion “CRLs 34 level was further confirmed in human CC cell lines and tissues.”

INTRODUCTION

1.     1.         The authors have mentioned CEA, CA125, and CA19-9 as the most used tumor biomarkers in clinical practice, but these are serum biomarkers clinicians are just checking the kinetics level of it to get an idea about tumor evolution and patient fellow up; recently there are many molecular biomarkers, and techniques using DNA mutation and RNA expression has been approved in CC prognosis, diagnosis, and management that authors may cite them in their introduction which is more related to their study.

2.     There are no scientifically precise links between the different paragraphs of the introduction to make a strong state of art authors may find a better relationship between the different parts of their introduction.

3.     The last paragraph must be reorganized usually this paragraph is to shed the light on the principal objective and aims of the study, but the authors are given a short description of their significant findings.

4.      The English can be improved.

MATERIALS AND METHODS

1.     In the Cell Culture section, how many generations of each cell line type were conducted, also at which generation these cells were used and how old the used flasks in the study.

2.     In the RNA Extraction and Quantitative Polymerase Chain Reaction (qPCR) the IRB approval number must be added.

3.     In the Cell Culture section, how many generations of each cell line type were conducted, also at which generation these cells were used and how old the used flasks in the study.

4.     The kit used for RNA extraction and reverse transcription reactions must be added with the company information (reference number, city, and country).

5.     The type of fluorescence used in the primers, the cDNA standardized concentrations, and the number of cycles of the normalized qPCR must be added to the body manuscript.

6.     More details for multi-factored model analysis must be added.

7.     Did any regression analysis was performed in the multi-factor analysis? If not, a regression analysis must be added to refine the survival results.

RESULTS

1.     The quality of Figure 1 must be improved and moved to the Materials and Methods section as it presents the workflow of the methodology utilized for the study. 

2.     This sentence is methods shouldn’t appear in the results section” The flow chart of the methodology utilized for our study is shown in Figure 1. The data for 458 COAD samples of 446 CC patients were downloaded from TCGA database (https://portal.gdc.Cancer.gov/ repository).

3.     Authors must add the p values of each variable to the text in this sentence to better guide the reader “The regression analysis results revealed that CRLs-risk score, age, and tumor stage were independent prognostic indictors for CC patients (Figure 4B−C).”

4.     The authors have used the adjacent peritumoral tissue and not a normal tissue in this sentence they must switch the word normal tissues to a peritumoral tissue as a control “Furthermore, the expression levels of 10 prognosis-associated CRLs were evaluated based on 20 paired CC and adjacent normal tissue samples from our institution”.

5.     Also, here “the corresponding matched-normal samples” must switch the normal samples to peritumoral control tissue.

6.     There are many typos and spelling mistakes in the results section that must be corrected and improved.

DISCUSSION

1.     The authors must give scientific information related to reference 2 like the classification of colon cancer among other cancers and the incidence of new cases with the estimated death in this sentence “Colon cancer is one of the most common malignancies worldwide in terms of both new cases as well as deaths, which seriously endangers human health [2]”.

2.     If authors are interested in the serum markers which, I think are not related to the purpose of their study they must include CA19-9 and CA125 to CEA as they showed more specificity for CC monitoring.

3.     3.         The authors are not discussing their proposed model with previous founding; do this model itself newly used in colon cancer but was it used for different kinds of cancer or is like a similar model if so authors must compare their CC results to other cancer models, and if not it has to be mentioned there is no such model was proposed in CC or neither in different kinds of cancer. 

4.     English could be improved poor.

CONCLUSION

Correct and clear.

Author Response

We thank you for your comments on our article. We have revised it according to your comments. Please see the attachment for more details.

Responses to the Comments from Reviewer 1

  1. Authors have constructed in silico their prognosis model; Their title must include that the model was constructed in silico and validated in cell lines and patient tissues.

Response: We thank the reviewer for this comment. Based on the reviewer’s suggestion, we included this part in the abstract. If we indicate in the title that the model was constructed in silico and validated in cell lines and patient tissues, the title would be too tedious for readers. Therefore we have made the changes accordingly in the abstract part, as follows:

Abstract:

Line 27−29: “Least absolute shrinkage and selection operator were utilized to construct the CRLs-based prognostic signature in silico for CC patients. CRLs level was validated in human CC cell lines and patient tissues.”

  1. The abstract's first sentence, "Colon cancer (CC) is one of the most common malignancies, which demand reliable prognostic biomarkers” must be more informative and scientifically address the health issues caused by colorectal cancer.

Response: We thank the reviewer for the comment. We have made the changes accordingly, as follows:

Abstract:

Line 21−23: “Colon cancer (CC) is one of the most common (6%) malignancies and leading cause of cancer-associated death (more than 0.5 million) worldwide, which demand reliable prognostic biomarkers.”

  1. The different parts of the abstract must be equiprobable in size, the methods part is longer than the others.

Response: We thank the reviewer for the comment. We have made the changes accordingly, as follows:

Abstract:

Line 26−29: “Data of CC patients were downloaded from public databases. The prognosis-associated CRLs were identified by co-expression analysis and univariate Cox. Least absolute shrinkage and selection operator were utilized to construct the CRLs-based prognostic signature in silico for CC patients. CRLs level was validated in human CC cell lines and patient tissues.”

  1. This sentence is part of the study methods and must be moved to the methods section, not between results and conclusion “CRLs 34 level was further confirmed in human CC cell lines and tissues.

Response: We thank the reviewer for the comment. We have made the changes accordingly, as follows:

Abstract:

Line 29: “CRLs level was validated in human CC cell lines and patient tissues.”

Line 33−34: “The prognostic prediction power of CRLs-risk score was further confirmed by cell lines, tissues and two independent CC cohorts.”

  1. The authors have mentioned CEA, CA125, and CA19-9 as the most used tumor biomarkers in clinical practice, but these are serum biomarkers clinicians are just checking the kinetics level of it to get an idea about tumor evolution and patient fellow up; recently there are many molecular biomarkers, and techniques using DNA mutation and RNA expression has been approved in CC prognosis, diagnosis, and management that authors may cite them in their introduction which is more related to their study.

Response: We thank the reviewer for the comment. We have made the changes accordingly, as follows:

Introduction:

Line 50−54: “On the other hand, although some new biomarkers such as circulating tumor DNA (ctDNA) have been approved in the prognosis, diagnosis and management of CC, they are also too expensive to be widely applied in clinical practice [13,14]. Therefore, de-veloping a reliable and practical prognostic signature is extremely important for the survival of CC patients.”

  1. There are no scientifically precise links between the different paragraphs of the introduction to make a strong state of art authors may find a better relationship between the different parts of their introduction.

Response: We thank the reviewer for the comment. We have made the changes accordingly, as follows:

Introduction:

Line 63−65: “In many studies, lncRNAs have also been utilized to construct prognostic models for various cancer types, including liver cancer and lung cancer [23, 24].”

  1. The last paragraph must be reorganized usually this paragraph is to shed the light on the principal objective and aims of the study, but the authors are given a short description of their significant findings.

Response: We appreciate the reviewer’s comments. We have made the changes accordingly, as follows:

Introduction:

Line 77−79: “In this study, we identified and validated a ten-CRLs-based prognostic model in CC patients based on data from TCGA. This model can be potentially applied in prognostic prediction and target therapy guidance in CC patients.”

  1. The English can be improved.

Response: We thank the reviewer for this comment. Based on your advice, we have improved the language quality of this manuscript with the help of a native speaker, Dr. Surendra Shukla from University of Oklahoma Health Sciences Center. The revised part has been marked Yellow.

  1. In the Cell Culture section, how many generations of each cell line type were conducted, also at which generation these cells were used and how old the used flasks in the study.

Response: We thank the reviewer for the comment. We have made the changes accordingly, as follows:

Materials and Methods:

Line 155−157: “HIEC-6, SW48 and Widr were used in the sixth generation, RKO, HCT8 and HCT116 were used in the eighth generation. All of cell lines were cultured in 60 mm TC-treated disposable Petri dish (430166, Corning®, New York, USA).”

  1. In the RNA Extraction and Quantitative Polymerase Chain Reaction (qPCR) the IRB approval number must be added.

Response: We thank the reviewer for the comment. We have made the changes accordingly, as follows:

Materials and Methods:

Line 164−166: “This study was approved by the ethics committee of the Sixth Affiliated Hospital of Sun Yat-sen University (IRB: 2020ZSLYEC-064, 04/20/2020).”

  1. In the Cell Culture section, how many generations of each cell line type were conducted, also at which generation these cells were used and how old the used flasks in the study.

Response: We thank the reviewer for the comment. We have made the changes accordingly, as follows:

Materials and Methods:

Line 155−157: “HIEC-6, SW48 and Widr were used in the sixth generation, RKO, HCT8 and HCT116 were used in the eighth generation. All of cell lines were cultured in 60 mm TC-treated disposable Petri dish (430166, Corning®, New York, USA).”

  1. The kit used for RNA extraction and reverse transcription reactions must be added with the company information (reference number, city, and country).

Response: We thank the reviewer for the comment. We have made the changes accordingly, as follows:

Materials and Methods:

Line 166−169: “TRIzol (15596018, Invitrogen, Carlsbad, USA) was utilized to collect total RNA samples from CC cell lines and paired CC and normal tissue specimens. ReverTra Ace quantitative polymerase chain reaction (qPCR) RT Kit (FSQ-101, Toyobo, Osaka, Japan) was used to perform the reverse transcription reaction.”

  1. The type of fluorescence used in the primers, the cDNA standardized concentrations, and the number of cycles of the normalized qPCR must be added to the body manuscript.

Response: We thank the reviewer for the comment. We have made the changes accordingly, as follows:

Materials and Methods:

Line 169−172: “SYBR Green (QP002, ESscience, Shanghai, China) was used as fluorescence. The quantitative PCR was conducted 40 cycles using 10ng/μl cDNA by ABI QuantStudio™ 7 Flex Real Time PCR Systems.”

  1. More details for multi-factored model analysis must be added.

Response: We thank the reviewer for the comment. After the cuproptosis-related lncRNAs screened out by univariate COX analysis, the overfitting lncRNAs would be filtered by LASSO regression. The rest of lncRNAs would be given a regression coefficient by the COXPH () function of “survival” R package to form a formula for the multi-factor COX regression analysis and to be performed collinearity test. Every patient would get a risk score according to the formula and their lncRNAs level. The prognostic model would be adopted when the P value of high risk and low risk groups’ KM curve was less than 0.05. Multivariate COX regression analysis was applied to find independent prognostic factors (P value), HR and 95% CI. The details of collinearity test were presented as follows:

(Figure 1)

(Figure 1). Collinearity Test: Generally, multiple collinearity is judged according to “variance inflation factor (VIF)” or “Tolerance”. The data is regarded as collinearity, when VIF > 10 or Tolerance < 0.1.

  1. Did any regression analysis was performed in the multi-factor analysis? If not, a regression analysis must be added to refine the survival results.

Response: We appreciate the reviewer’s comments. After inspection and confirmation again, regression analysis had been performed in the multi-factor COX regression analysis. The details were presented as follows:

(Figure 2)

A                                                             

B

 (Figure 2). (A) Operation code of regression analysis. (B) Regression coefficient of 10 cuproptosis-related lncRNAs.

  1. The quality of Figure 1 must be improved and moved to the Materials and Methods section as it presents the workflow of the methodology utilized for the study.

Response: We thank the reviewer for the comment. We have made the changes accordingly, as follows:

Materials and Methods:

Line 105−107: “ExplncRNAi is the expression value of lncRNA, and CoeflncRNAi is the estimated regression coefficient of indicated lncRNA in multivariable Cox regression analysis (see Figure 1).”

Line 117−118: “Moreover, a nomogram based on CRLs signature and clinical characteristics was es-tablished to predict survival of CC patients (see Figure 1).”

(Figure 3)

(Figure 3). Figure 1. Flow Chart of This Research.

  1. This sentence is methods shouldn’t appear in the results section” The flow chart of the methodology utilized for our study is shown in Figure 1. The data for 458 COAD samples of 446 CC patients were downloaded from TCGA database (https://portal.gdc.Cancer.gov/ repository).

Response: We thank the reviewer for the comment. We have removed above sentences from the result part. We have made the changes accordingly, as follows:

Results:

Line 176−177: “The flow chart of methodology utilized for our study is shown in Figure 1. Based on the data from TCGA-COAD, a total of 16,876 lncRNAs were identified.”

  1. Authors must add the p values of each variable to the text in this sentence to better guide the reader “The regression analysis results revealed that CRLs-risk score, age, and tumor stage were independent prognostic indictors for CC patients (Figure 4B−C).

Response: We thank the reviewer for the comment. We have made the changes accordingly, as follows:

Results:

Line 227−229: “The regression analysis results revealed that CRLs-risk score (P < 0.001), age (P < 0.001), and tumor stage (P = 0.004) were independent prognostic indictors for CC patients (Figure 4B−C).”

  1. The authors have used the adjacent peritumoral tissue and not a normal tissue in this sentence they must switch the word normal tissues to a peritumoral tissue as a control “Furthermore, the expression levels of 10 prognosis-associated CRLs were evaluated based on 20 paired CC and adjacent normal tissue samples from our institution”.

Response: We thank the reviewer for the comment. We have made the changes accordingly, as follows:

Results:

Line 320−322: “Furthermore, the expression levels of 10 prognosis-associated CRLs were evaluated based on 20 paired CC and peritumoral tissue samples from our institution.”

  1. Also, here “the corresponding matched-normal samples” must switch the normal samples to peritumoral control tissue.

Response: We thank the reviewer for the comment. We have made the changes accordingly, as follows:

Results:

Line 324−327: “Conversely, AL031985.3, FARSA-AS1, LINC01762, PDE9A-AS1, AC104964.3, and AC092375.2 showed lower expression levels in tumor tissues than the corresponding matched-peritumoral samples (Figure 9B).”

(Figure 4)

(Figure 4). Figure 9B: Expression level of ten prognosis-associated CRLs in 20 paired CC and peritumoral tissue samples.

  1. There are many typos and spelling mistakes in the results section that must be corrected and improved.

Response: We thank the reviewer for this comment. Based on the reviewer’s advice, we have corrected mistakes and improved the language quality of this manuscript with the help of a native speaker, Dr. Surendra Shukla from University of Oklahoma Health Sciences Center. The revised part has been marked Yellow, some of them as follow:

Results:

Line 181−182: “Sankey diagram was established to demonstrate the relationship between CRLs and CRGs in CC patients (Figure 2A).”

Line 184−188: “Then LASSO regression analysis was performed to filter the overfitting CRLs, which identified 10 prognosis-associated CRLs (LINC02257, AL031985.3, FARSA-AS1, LINC01762, PDE9A-AS1, NSMCE1-DT, AC002066.1, AC104964.3, AC010789.2 and AC092375.2) by multivariate Cox regression analysis (Figure 2C−D).”

Line 213−214: “Moreover, the expression levels of 10 prognosis-associated CRLs were presented in the heatmap (Figure 3C).”

Line 229−232: “Interestingly, the CRLs-risk score in CC patients with lymphatic invasion (LI) was significantly higher than those without LI, similar result was also observed in CC patients with venous invasion (VI) (Figure 4D−E).”

  1. The authors must give scientific information related to reference 2 like the classification of colon cancer among other cancers and the incidence of new cases with the estimated death in this sentence “Colon cancer is one of the most common malignancies worldwide in terms of both new cases as well as deaths, which seriously endangers human health [2]”.

Response: We thank the reviewer for the comment. We have made the changes accordingly, as follows:

Discussion:

Line 353−355: “Colon cancer is one of the most common (6%) malignancies worldwide in terms of both new cases (more than 1.14 million) as well as deaths (more than 0.57 million), which seriously endangers human health [2].”

  1. If authors are interested in the serum markers which, I think are not related to the purpose of their study they must include CA19-9 and CA125 to CEA as they showed more specificity for CC monitoring.

Response: We thank the reviewer for this comment. Based on the reviewer’s advice, we deleted relevant content. We have made the changes accordingly, as follows:

   Results:

Line 236−237: “More importantly, the AUC value of CRLs-risk score (0.712) was higher than that of age (0.621), tumor stage (0.617) and gender (0.480) (Figure 5B).”

Discussion:

Line 355−359: “Although the detection and treatment approach have been greatly improved, the 5-year survival rate is only around 64% for CC patients and this data is less than 20% for CC patients with distant metastasis [7, 43]. Therefore, identifying the stable and effective prognostic predictors is warranted to improve the survival of CC patients.”

Line 368−370: “CRLs signature was demonstrated with powerful and accurate prediction power in TCGA-COAD cohort, which could better guide the clinical management of CC patients.”

(Figure 5)

(Figure 5). Figure 5B: ROC curve of the CRLs-risk score and clinical characteristics for prognostic prediction power in CC patients.

  1. The authors are not discussing their proposed model with previous founding; do this model itself newly used in colon cancer but was it used for different kinds of cancer or is like a similar model if so authors must compare their CC results to other cancer models, and if not it has to be mentioned there is no such model was proposed in CC or neither in different kinds of cancer.

Response: We thank the reviewer for the comment. We have added the study about cuproptosis-related-lncRNAs prognostic model of other cancer types in our discussion part, as follows:

Discussion:

Line 374−379: “Compared with CRLs signature in other cancer types, such as five-CRLs signature (FOXD2-AS1, NRAV, MED8-AS1, WARS2-AS1 and MKLN1-AS) in hepatocellular carcinoma and three-CRLs signature (AC026401.3, FOXD2-AS1, and LASTR) in Clear Cell Renal Cell Carcinoma [44, 45], our study constructed a prognostic signature based on 10 CRLs (LINC02257, AC002066.1, NSMCE1-DT, AC010789.2, AL031985.3, FARSA-AS1, LINC01762, PDE9A-AS1, AC104964.3, AC092375.2) for CC patients.”

  1. English could be improved poor.

Response: We thank the reviewer for this comment. Based on your advice, we have improved the language quality of this manuscript with the help of a native speaker, Dr. Surendra Shukla from University of Oklahoma Health Sciences Center. The revised part has been marked Yellow.

Reviewer 2 Report

In this manuscript by Liang et al, a CRLs-based prognostic model, which could be potentially used in prognostic prediction and as a guidance for targeted therapy-selection of CC patients, is provided. I think this work is very interesting because there are still few information on cuproptosis and cancer. The study is really well executed and the experiments of validations performed on CC patients (Fig 9) make all the data very convincing.

Minor points:

-The authors claimed in high CRLs-risk group a high expression of LINC02257, AC002066.1, NSMCE1-DT and AC010789.2 but and expression of AL031985.3, FARSA-AS1, LINC01762, PDE9A-AS1, AC104964.3, and 211 AC092375.2. These lncRNAs different expression is also demonstrated in CC patients (Fig 9). To further increase the impact of their work, the authors should demonstrate the implication of some of these non-coding RNAs in CC-related pathways (different from those already mentioned in the discussion).

-In the first part of the results the authors affirm: “In addition, survival analysis results illustrated that CRLs-risk score exhibited a better prognostic prediction power in stage I-II CC patients than stage III and stage IV patients”. This result is little discussed later. Can the authors better discuss this finding? Why do they think the CRLs-risk score presented better prediction efficacy for CC stage I-II ?

-Please correct fig 6D with “Tumor mutation burden”. 

Author Response

We thank you for your comments on our article. We have revised it according to your comments. Please see the attachment for more details.

Responses to the Comments from Reviewer 2

  1. The authors claimed in high CRLs-risk group a high expression of LINC02257, AC002066.1, NSMCE1-DT and AC010789.2 but and expression of AL031985.3, FARSA-AS1, LINC01762, PDE9A-AS1, AC104964.3, and 211 AC092375.2. These lncRNAs different expression is also demonstrated in CC patients (Fig 9). To further increase the impact of their work, the authors should demonstrate the implication of some of these non-coding RNAs in CC-related pathways (different from those already mentioned in the discussion).

Response: We appreciate the reviewer’s comments. We have added the study about lncRNA NSMCE1-DT in regulating prognosis of colon cancer patients in our discussion part. More molecular mechanisms for these 10 cuproptosis-related lncRNAs warrant further investigations in future work. We have made the changes accordingly, as follows:

Discussion:

Line 390−392: “In addition, NSMCE1-DT, AL031985.3 and LINC01762 were reported to be involved in the prognosis of colon adenocarcinoma, hepatocellular carcinoma and renal cell carcinoma respectively [50−52].”

  1. In the first part of the results the authors affirm: “In addition, survival analysis results illustrated that CRLs-risk score exhibited a better prognostic prediction power in stage I-II CC patients than stage III and stage IV patients”. This result is little discussed later. Can the authors better discuss this finding? Why do they think the CRLs-risk score presented better prediction efficacy for CC stage I-II ?

Response: We appreciate the reviewer’s comments. The subgroup survival analysis illustrated that CRLs-risk score exhibited a better prognostic prediction power in stage I-II CC patients than stage III and stage IV patients. These 10 CRLs may play an important role during the onset and early development of CC, which can explain above phenomenon. The underlying molecular mechanisms for these CRLs in regulating CC onset and early development need to be explored in our future work. We have made the changes accordingly, as follows:

Discussion:

Line 360−365: “The subgroup survival analysis illustrated that CRLs-risk score exhibited a better prognostic prediction power in stage I-II CC patients than stage III and stage IV patients. These 10 CRLs may play an important role during the onset and early development of CC, which can explain above phenomenon. The underlying molecular mechanisms for these CRLs in regulating CC onset and early development need to be explored in our future work.”

  1. Please correct fig 6D with “Tumor mutation burden”.

Response: We thank the reviewer for pointing out this typo. We have corrected the Figure 6D accordingly, as follows:

Results:

(Figure 1)

(Figure 1). Figure 6D: Violin plot for statistical analysis of TMB in high CRLs-risk and low CRLs-risk CC patients (P = 0.039).
